# MicroRNAs Are Involved in Regulating Plant Development and Stress Response through Fine-Tuning of TIR1/AFB-Dependent Auxin Signaling

**DOI:** 10.3390/ijms23010510

**Published:** 2022-01-03

**Authors:** Pan Luo, Dongwei Di, Lei Wu, Jiangwei Yang, Yufang Lu, Weiming Shi

**Affiliations:** 1College of Life Science and Technology, Gansu Agricultural University, Lanzhou 730070, China; yjw@gsau.edu.cn; 2State Key Laboratory of Soil and Sustainable Agriculture, Institute of Soil Science, Chinese Academy of Sciences, Nanjing 210008, China; yflu@issas.ac.cn (Y.L.); wmshi@issas.ac.cn (W.S.); 3MOE Key Laboratory of Cell Activities and Stress Adaptations, School of Life Sciences, Lanzhou University, Lanzhou 730000, China; leiwu@lzu.edu.cn

**Keywords:** auxin signal, TIR1/AFBs (Transport Inhibitor Response 1/Auxin Signaling F-Box Protein), microRNA, precise regulation

## Abstract

Auxin, primarily indole-3-acetic acid (IAA), is a versatile signal molecule that regulates many aspects of plant growth, development, and stress response. Recently, microRNAs (miRNAs), a type of short non-coding RNA, have emerged as master regulators of the auxin response pathways by affecting auxin homeostasis and perception in plants. The combination of these miRNAs and the autoregulation of the auxin signaling pathways, as well as the interaction with other hormones, creates a regulatory network that controls the level of auxin perception and signal transduction to maintain signaling homeostasis. In this review, we will detail the miRNAs involved in auxin signaling to illustrate its in planta complex regulation.

## 1. Introduction

The plant growth regulator, auxin, participates in nearly all aspects of plant growth and development, including cell expansion, vascular differentiation, lateral root (LR) formation, hypocotyl elongation, senescence, abscission, hormone crosstalk and stress responses [1,2,3,4]. The most common natural auxin is indole-3-acetic acid (IAA), which can perform most of the regulatory functions of auxins in plants [5]. In addition, indole-3-butyric acid (IBA), 4-chloroindole-3-acetic acid (4-Cl-IAA), and phenylacetic acid (PAA) have been also detected in plants [6,7]. One of the most fascinating questions in auxin biology is how such a simple signal molecule can exhibit distinct functions in plants [8]. 

Auxin is mainly perceived by a co-receptor complex that consists of two protein families, the Transport Inhibitor Response 1/Auxin Signaling F-box Protein (TIR1/AFB) family and the Auxin/Indole Acetic Acid (Aux/IAA) family. After auxin perception by the SCF^TIR1/AFB^-Aux/IAA complex, the transcription of downstream genes is regulated directly by a third protein family, the Auxin Response Factor (ARF) family [8]. Auxin acts as an adaptor that determines which protein will be bound with an Aux/IAA transcriptional repressor. In the absence of auxin, the Aux/IAAs bind to the ARF transcriptional factors to inhibit ARF mediated auxin-responsive genes transcription; however, when auxin is present, it mediates the binding of Aux/IAA proteins to SCF^TIR1/AFB^ to form a SCF^TIR1/AFB^-Aux/IAA complex, which leads to the ubiquitination and degradation of Aux/IAA by the 26S proteasome and frees the ARFs to regulate up or down their target genes [9]. Interestingly, auxin cannot bind the TIR1 directly without an Aux/IAA [9]. Many auxin-responsive genes fall into one of three major families: SAUR (Small auxin up RNA), Aux/IAA, and GH3 (Gretchen Hagen 3) [10,11]. 

In addition to the precise regulation of auxin biosynthesis and distribution, the auxin signaling pathway is subject to locus-specific control by microRNAs (miRNA) [12,13]. The miRNAs are short, single-stranded nucleic acid molecules (~21–24 nucleotides) that repress target gene expression as an additional layer of regulation of plant growth, and stress responses in plants [14]. In plants, primary miRNAs (pri-miRNAs) are endogenously expressed from miRNA-encoding genes (MIRs) by the DNA-dependent RNA polymerase II (Pol II) [15]. The pri-miRNA is cleaved by RNase III enzymes Dicer-Like 1(DCL1)/DCL4 and then processed into a mature miRNA by a protein complex composed of Serrate (SE), Hyponastic Leaves 1 (HY1), Touch and DCL1 [16]. The miRNA biogenesis is also regulated by various protein factors at the transcriptional and posttranscriptional levels [17]. A recent study reports a new protein factor, SE-Associated Protein 1 (SEAP1), that associates with the DCL1 complex and promotes the interaction of the DCL1 complexes with pri-miRNAs, facilitating pri-miRNA splicing, processing, and/or stability [17]. The miRNAs regulate gene expression by directly cleaving target mRNA or inhibiting translation in the cytoplasm through conserved Argonaute (AGO) proteins [16]. In addition, miRNAs can bind to target promoters in the nucleus, thereby regulating expression [18], suggesting that miRNAs can function as a gene activator or repressor. In plants, miRNAs exhibit high evolutionarily conserved complementarity to mRNAs among dicot and monocot species [19]. Conserved miRNAs may play important roles in basic development and signal transduction, while non-conserved miRNAs may play particular roles in different tissues or plant species under special environmental conditions [20,21,22].

The genes involved in auxin signal transduction are the targets of multiple miRNAs, with some genes extensively studied in relation to plant growth and stress responses. The miRNA-mediated regulation of auxin signaling is generally divided into two types: (i) by direct targeting of the genes involved in the auxin signaling pathway, including TIR1/AFBs, ARF and AUX/IAA; or (ii) by modulating free auxin content via indirect regulation of auxin biosynthesis, metabolism and transport. Here, we will mainly review the latest studies on miRNA-mediated regulation of the auxin pathway in the dicot Arabidopsis (*Arabidopsis thaliana*) and the monocot rice (*Oryza sativa*), as well as some new findings in other species, in order to provide a comprehensive picture of the role of miRNAs in auxin signaling and how they finely regulate development and stress responses in the plant kingdom.

## 2. miRNA-Mediated Regulation of Auxin Signaling and Homeostasis

### 2.1. Two Conserved miRNAs, miR160 and miR167, Constitute a Complex Feedback Loop to Regulate Auxin Signaling and Homeostasis

The miRNAs, miR160 and miR167, are two conserved regulators of the auxin signaling pathway. They modulate the expression of several *AUXIN RESPONSE FACTOR* (*ARF*) genes, which function as activators or repressors of primary auxin responsive genes in plants [23,24,25]. In Arabidopsis, miR160 directly cleaves the mRNA of *AtARF10/16/17*, while miR167 cleaves *AtARF6/8* [26,27,28,29,30]. Plants expressing miRNA-resistant forms of these *ARFs* exhibit pleiotropic developmental defects, indicating the essential roles of miR160 and miR167 in Arabidopsis growth and development [27,28]. Surprisingly, there are multiple feedback points in the miR160-miR167/ARF regulatory network.

miR160/167 and *AtARF6/8/17* constitute a regulatory network that controls adventitious rooting [26]. In this regulatory network, miR160/ARF17 is a negative regulator of adventitious rooting, while miR167/ARF6/8 is a positive regulator. These three ARFs precisely control their own expression at both the transcriptional and posttranscriptional levels by regulating miR160 and miR167 availability, while miR160 and miR167 homeostasis also provides feedback control by AtARF6/8/17 [26] (Figure 1A). Moreover, the expression of miRNA-resistant *AtARF17* in transgenic Arabidopsis plants (mAtARF17) increases the level of *AtARF17*, which alters the accumulation of the auxin-conjugating genes *AtGH3.2/3.3/3.5/3.6* and results in pleiotropic developmental defects [28], suggesting that the miR160/ARF17 module is more critical for proper development via regulation of IAA homeostasis.

In addition to the control of root growth, the miR160/ARF10/16/ABSCISIC ACID INSENSITIVE 3 (ABI3) regulatory loop is also essential for several opposite processes, including seed dormancy, seed germination, and somatic embryogenesis [27,31,32,33]. Firstly, auxin can enhance ABA-mediated seed dormancy by employing *AtARF10/16*, the target genes of miR160, to maintain the transcription of *AtABI3* [32]. Secondly, AtABI3 inhibits the transcription of miRNA-encoding gene *AtMIR160b* and then the decreased miR160 accumulation. As a result, the decreased miR160 post-transcriptionally regulates AtARF10/16, while AtARF10/16 may regulate the auxin level through AtLEC2-mediated *AtYUC1/4/10* activation, which finally controls somatic embryogenesis [31,32,33,34]. Thirdly, the miR160/ARF10 module is also involved in regulating seed germination under different concentrations of ABA [27] (Figure 1A). Together, these results clearly suggest that the auxin and ABA signaling pathways synergically control seed dormancy and germination via the miR160/ARF10/16/ABI3 regulatory loop. Moreover, miR167 controls both male and female reproduction via regulating the expression patterns of *AtARF6* and *AtARF8* [30]. However, the environmental or developmental signals and the mechanisms underlying the transition of seed dormancy/germination and somatic embryogenesis still need to be further investigated.

Furthermore, miR160 and miR167 also respond to biotic or abiotic stresses (Figure 2A). The miR160 prompts the expression of heat shock proteins and plant development via reducing its targets, *AtARF10/16/17*, to allow plants to survive heat stress [35]. The miR167 is involved in the response to high osmotic stress by decreasing Indoleacetic Acid Alanine-Resistant3 (AtIAR3), a hydrolytic enzyme that generates an active form of auxin, but not *AtARF6/8* [36]. In contrast, AtARF6/8 promotes jasmonic acid (JA) production [37], while JA induces the transcription of *AtIAR3* (also named as *JASMONATE RESPONSIVE3*, *JR3*). The active IAA that is released by AtIAR3 also stimulates the expression of *AtARF6/8*. Thus, *AtARF6/8* may be indirectly involved in the response to osmotic stress [36]. The miR167 also modulates the defense against pathogens through *AtARF6**/F8*. Upon infection by the bacterial pathogen *Pseudomonas syringae*, overexpression of *miR167* decreases *At**ARF6/8* levels and the size of stomatal apertures, which can reduce pathogen entry into leaves in cooperation with salicylic acid (SA) [38].

In rice, OsmiR160 has four potential targets similar to *AtARF10* and *AtARF16*, namely, *OsARF8*, *OsARF10*, *OsARF18*, and *OsARF22* [39]. Of them, OsARF18 has the highest similarity to AtARF16. Rice plants expressing an OsmiR160-resistant version of *OsARF8* show pleiotropic defects in growth and development, including reduced starch accumulation, dwarf stature and small seeds [39]. Interestingly, the expression of MIR160a and MIR160b is decreased in mOsARF18 plants, which expresses an miR160-resistant version of *OsARF18* in transgenic plants, indicating the negative regulation of OsARF18 by OsmiR160 [39] (Figure 1A). OsmiR167 also has four target *OsARFs* (*OsARF6*, *OsARF12*, *OsARF17* and *OsARF25*). The OsmiR167-OsARF8-OsGH3.2 pathway functions in response to exogenous auxin [40]. OsmiR167-guided OsARF12 expression regulates root elongation and iron accumulation in rice [41], while OsmiR167a represses its targets, OsARF12/17/25, to control rice tiller angle by modulating the asymmetric auxin distribution in shoots [42]. Recently, the OsmiR167a-OsARF6-OsAUX3 module has been reported to regulate grain length and weight in rice, indicating the OsmiR167a-mediated auxin distribution may be important for rice yield improvement [43]. Overall, miR160 and miR167 may modulate auxin signaling not only through the simple cleavage of ARFs, but also through influencing auxin homeostasis and response during specific developmental processes (Figure 2A).

In addition to Arabidopsis and rice, the roles of miR160 and miR167 in regulating plant growth and stress responses have also been widely investigated in multiple different species, including wheat, maize, and barley (Table 1). Moreover, the vital role of auxin signaling in nodule initiation, primordia development and nodule organogenesis has been systematically investigated [44,45]. Formation of the root nodule in legumes is also regulated by miR160 and miR167 (Table 1). Consistent with the opposing roles of miR160 and miR167 in regulating adventitious root development in Arabidopsis, these two miRNAs function antagonistically to regulate nodule formation and root system architecture in soybean (*Glycine max*) [46].

Overall, the miR160-miR167-ARF module represents a critical regulatory node that adjusts a wide range of processes in response to different environmental and developmental stimuli, including embryonic development, root initiation, and reproductive organ maturation.

### 2.2. The miR390-TAS3-tasiARF Module Is a Regulatory Hub of Auxin Signaling

In plants, miR390 directs the production of *trans*-acting small interfering RNAs (tasiRNAs) from the mRNA of *trans-acting*
*Sirna 3* (*TAS3*) to downregulate ARF2/3/4, which are known as tasiARFs [47,48]. The miR390/TAS3/ARF system is evolutionarily conserved [48], and miR390 functions as a regulatory hub in the auxin signaling pathway through: (i) transcription of *MIR390a* and *MIR390b* induced by exogenous auxin in a dose-dependent manner in seedling roots [49]; (ii) miR390 initiating the biosynthesis of tasiARFs from TAS3 mRNA, which then cleaves *AtARF2/3/4* mRNA [50,51,52]; (iii) ARF3 and ARF4 positively or negatively regulating the accumulation of *MIR390a* [51]; and (iv) AtARF5 directly binding to the auxin response elements (AuxREs) of the *MIR390* promoter and positively regulating its transcription [53] (Figure 1B). In roots, miR390 is specifically expressed at the sites of lateral root (LR) initiation and the transit-amplifying compartment of the meristem, where it controls the spatiotemporal expression of tasiRNAs to prompt LR emergence and response to exogenous IAA in the meristem [49,51,53,54]. Impaired tasiARF production results in decreased young LR primordia (stages 1 to 4), while the elevated tasiARFs exhibit increased LR primordia (stages 5 to 7) [51]. In leaves, miR390-derrived tasiARFs move intercellularly to create a small tasiARFs gradient and pattern the abaxial determinant *AtARF3.* Subsequently, AtARF3 regulates the expression of downstream genes involved in leaf morphogenesis and developmental timing and patterning [51,55,56]. Considering the potential ability of tasiRNAs to move in a non-cell autonomously manner, the miR390-TAS3-tasiARF module may function as a communication signal to determine the spatial expression pattern of AtARF2/3/4 and to precisely control cell elongation and differentiation during organogenesis in response to different environmental conditions.

As a conserved miRNA in plants, the functions of miR390-mediated regulation during plant development, biotic and abiotic stresses have been investigated in different species (Table 1). Interestingly, miR390 plays dual roles in regulating nodulation and in response to salt stress. In *Medicago truncatula*, overexpression of miR390 results in promoting LR growth but preventing nodule organogenesis, rhizobial infection, and inhibiting the transcription of nodulation genes, while inactivation of the miR390/TAS3 module increases nodulation and rhizobial infection [57]. In *Helianthus tuberosus*, 100 mM of NaCl induces the expression of miR390, while 300 mM of NaCl inhibits the expression of miR390, which alters the salt tolerance [58]. These distinct roles of miR390 in regulating nodulation and salt stress also indicates the regulatory complexity of the miR390-TAS3-tasiARFs module.

In addition, the expression of miR390 is regulated by AtARF3/4/5. Among these ARFs, AtARF5 is not a target of the miR390-TAS3-tasiARF module, suggesting that the accumulation of miR390 is controlled by the auxin level and/or feedback regulation. This also implies that the miR390/TAS3/tasiARF module is a hub of the regulatory networks mediating the auxin response [53]. It would be interesting to investigate which ARFs are employed to control the expression of miR390 during different developmental contexts.

### 2.3. miR393, a Repressor of Auxin Signaling

Another conserved miRNA is miR393, which has been discovered in many plants (Table 1). In Arabidopsis, miR393 is encoded by two loci, *AtMIR393a* and *AtMIR393b*, and has four known F-box target genes, namely the TIR1/AFB Auxin Receptors 1–3 (TAAR1-3) [59,60]. The *tir1/afb* quadruple mutant exhibits severe developmental defects, indicating the key role of TAAR-dependent auxin signaling [61]. Previous studies have demonstrated that the miR393-TIR1/AFB regulatory module plays a critical role in root system architecture, leaf development, auxin response and plant response to abiotic and biotic stress (Figure 1C and Figure 2C) [12]. Under salt stress, NaCl induces miR393 expression and inhibits *TIR1/AFB2* via enhancing the transcription of miR393a, and then inhibits LR initiation, emergence and elongation (Figure 2C) [62]. The peptide flg22 derived from *Pseudomonas syringae* induces the transcription of *AtMIR393a* and then negatively regulates *TIR1/AFB2/3* in plants to confer transcriptional repression and decrease disease sensitivity (Figure 2C) [63]. In aerial organs, *AtMIR393b* is the predominant source for mature miR393, and miR393 can guide the cleavage of all four TAAR genes [64]. Surprisingly, miR393 also regulates TAAR expression by initiating secondary siRNA biogenesis of AtAFB2/3 (siTAAR) in aerial organs [64]. In addition, exogenous IAA promotes the accumulation of miR393 in aerial organs via inducing the transcription of *AtMIR393b*, but not *AtMIR393a* [65]. Overexpression of the miR393-resistant TIR1 (mTIR1) increases the transcription of miR393, indicating a positive regulation of AtTIR1 at the level of miR393 transcripts. Interestingly, a nitrate (NO_3_^−^) signal can directly upregulate the transcription of AtAFB3, although the metabolism of nitrate, ammonium (NH_4_^+^) and glutamine (Gln) positively regulate the transcription of miR393 (Figure 1C). This feed-forward mechanism can reset both miR393 and its target genes to basal levels [66]. Furthermore, *MIR393a* and *MIR393b* synergistically function in somatic embryogenesis and response to drought stress by negatively regulating the expression of *AtTIR1/AFB2* [67,68].

Similarly, in rice, miR393 is encoded by two loci, *OsMIR393a* and *OsMIR393b* [59,69], and the conserved OsTIR1 and OsAFB2 were identified as target genes of miR393 by degradome sequencing [70] and 5′-RACE [71]. Spatial expression analysis by GUS staining showed that OsMIR393a is strongly expressed in the crown and in adventitious roots, but not in the primary root, while OsMIR393b is expressed in the shoot apical meristem, coleoptile tips and stomata cells, but cannot be detected in the roots [72,73]. Overexpression of *OsMIR393a* and *OsMIR393b* caused flag leaf inclination, primary/crown root growth and seed germination defects in rice [72,73]. These data in Arabidopsis and rice suggest that the locus-specific regulation of MIR393 transcription provides another layer for modification of the auxin signaling pathway via cleaving the target genes.

Additionally, the roles of miR393 in regulating both normal plant developmental processes, such as nodulation, adventitious root formation, leaf morphogenesis and fruit development, and the response to different environmental stresses, including aluminum stress, drought stress, cold stress and aphid infection, have been widely studied in different species (Table 1). However, due to a lack of a knockout mutant of *MIR393*, the exact roles of miR393 in plant development and stress responses still need more investigation.

### 2.4. miR164 and miR169, Two Modulators of Auxin Signaling

The miR164 family is comprised of three members (miR164a/b/c) in Arabidopsis but six members (miR164a/b/c/d/e/f) in rice [112,113,114]. The miR164 molecules guide the cleavage of mRNAs encoding NAC transcription factors [112,115]. *AtNAC1* promotes LR development through transduction of the auxin signal downstream of AtTIR1 [115,116]. Overexpression of *AtNAC1* can restore LR formation in *Attir1*, and antisense of *AtNAC1* can block AtTIR1-induced LR development [115]. Exogenous IAA induces the transcription of AtMIR164, which disappears in the mutants *auxin resistant 1/2* (*axr1/2*) and *tir1* [116]. AtmiR164 also functions as a hub in regulating leaf differentiation and age-dependent cell death [113,117] and functions downstream of AtTCP3, cooperating with the auxin signaling components AtSAUR and AtIAA3 to inhibit the expression of AtCUC1/2 to regulate leaf differentiation [117]. The miR164, AtNAC2 and AtEIN2 form a small regulatory loop for age-dependent cell death in leaves, with the AtEIN2 inhibiting the transcription of *AtmiR164* and promoting that of *AtNAC2,* while AtmiR164 guides the cleavage of AtNAC2 mRNA [113] (Figure 1D). Furthermore, the conserved target genes of miR164 in rice, *OsNAC/MTN2/3/4/6*, are reported to negatively regulate the drought stress response via miR164-mediated cleavage (Figure 2D). In tomato, miR164 is also involved in the response to viral infection via cleavage of *SlNAC1* [79,97], although it remains unclear if this involves auxin signaling [112]. Taken together, miR164 may be a modulator in NAC-dependent auxin signaling, where auxin prompts the generation of miR164, which then can clear NAC mRNA to downregulate the auxin signal [116].

The fact that miR169 can respond to cold stress was found through a genome-wide expression analysis in wild-type and *solitary root 1* (*Atslr1*; a knockout of *Aux/IAA14*) [118]. Further gene expression analysis showed that AtIAA14 positively regulates the transcription of the miR169 precursor *AtMIR169* [118]. Moreover, the AtmiR169 target gene, *AtNF-YA2*, directly binds to the promoters of flowering time (FT) and the auxin biosynthesis gene *AtYUC2* in response to ambient temperature [119] (Figure 2E). Hence, the auxin-Aux/IAA14-miR169 module might constitute a simple regulatory loop to control the free IAA content and then auxin signal transduction in response to temperature stress (Figure 2E). It is therefore necessary to clarify the exact function of miR164 in temperature stress and/or developmental processes using an miR164 knockout mutant.

### 2.5. miR847, an Activator of Auxin Signaling

While miR847 is a low-abundance miRNA, and its accumulation cannot be detected under normal conditions [46], a rapid induction of miR847 is observed upon auxin treatment, and this induction disappears in *Attir1* and *Ataxr1-3* mutants, demonstrating that *miR847* functions downstream of AtTIR1 [46]. Deep sequencing results show that the miR847-coding gene, *AtMIR847*, is an orthologous conserved locus in the Arabidopsis lineage [120]. A 5′ RACE assay demonstrated the miR847-mediated cleavage of *AtIAA28* mRNA. Phenotypic analysis in lines overexpressing or carrying a mutation of miR847 and *AtIAA28* showed that miR847 positively regulates aerial lateral organ development, LR development, meristematic competence and cell proliferation by de-repressing the *AtIAA28*-dependent auxin signaling pathway (Figure 1E) [46]. The miR847-IAA28 module is concurrent with proteasome-dependent degradation of IAA28 in response to development-induced auxin content alteration and may offer a more precise switch to regulate the IAA28-dependent auxin signaling cascades.

### 2.6. miR156, miR165 and miR166, a New Mode of Action for Regulators of Auxin Homeostasis and Signaling

Except for the miRNAs that directly regulate the auxin signaling pathway as mentioned above, there are other miRNAs that directly regulate the auxin content and indirectly regulate auxin responses, e.g., miR156, miR165 and miR166 (Figure 1F,G). In rice, OsmiR156 guides the cleavage of *Squamosa Promoter Binding Protein-like 7* (*Os**SPL7*), while OsSPL7 can directly bind to the promoter of *OsGH3.8*, which catalyzes the ATP-dependent formation of IAA-amino acid conjugates [121,122]. Overexpression of *OsGH3.8* in miR156-overexpression (miR156f-OE) lines can partially complement the developmental defects observed in miR156OE plants, e.g., fewer tillers and increased plant height, indicating that miR156f/SPL7 can directly control OsGH3.8 to regulate plant growth and development (Figure 1F) [122]. In addition, miR165 and miR166 are also reported to regulate auxin accumulation through the activation of AtYUC1/4/10 by the transcription factors Phabulosa/Phavoluta (PHB/PHV) and Leafy Cotyledon2 (LEC2) in Arabidopsis somatic embryogenesis (Figure 1G) [34]. The transcription of *AtLEC2* is also upregulated by AtARF10/16, the targets of miR160, indicating a complex regulation of miR160, miR165 and miR166 in controlling somatic embryogenesis (Figure 1A,G). In addition, miR166 in *Larix leptolepis* is also reported to inhibit the transcription of *LaNIT* (an auxin biosynthetic gene) and *LaARF1/2* through LaHDZIPIII during somatic embyro germination (Table 1) [95]. More studies are still needed to clarify the role of these three miRNAs, including creating knockout mutants of the encoding gene.

### 2.7. Non-Conserved miRNA-Auxin Module in Plants

As mentioned above, the conserved miRNAs, including miR160, miR167, miR390 and miR393, have been extensively investigated in Arabidopsis, rice, tomato, maize, potato, and barley (Figure 1 and Figure 2; Table 1). In addition, there have been some non-conserved miRNAs reported to regulate auxin signaling. During post-mowing regeneration in winter wheat, miR1153-y induces the specific cleavage of *TaGH3.7*, indicating the potential role of miR1153-y in auxin signaling [95,110]. In litchi (*Litchi chinensis* Sonn.), a novel miRNA-ARF (miRN43-ARF9) regulatory pathway has been discovered, implying that species-specific miRNA may function in the auxin signaling pathway [111]. Another study shows that a long noncoding RNA (lncRNA354) functions as a competitor of GhmiR160b to regulate GhARF17/18 genes under salt stress in upland cotton, suggesting that endogenous RNA in other species may also play roles in regulating miRNA-mediated auxin signaling [87]. Taken together, further studies on different plant species grown in different environments are necessary to identify and investigate new miRNA-auxin signaling modules, which will extend our understanding of the regulation and function of miRNA and auxin signaling.

## 3. The Crosstalk of Auxin, miRNA and Other Hormones during Plant Development and Stress Responses

Auxin controls most developmental stages and stress responses in plants, and the communication between auxin and the other phytohormones (ethylene, abscisic acid, jasmonic acid, cytokinin, etc.) is critical to these processes. Although it is clear that there is phytohormone crosstalk, only a few mediators have been investigated thus far, including miRNAs [55].

The miRNA is involved in multiple developmental processes, including organs morphogenesis, root growth, leaf development, and stress responses, which is a characteristic common to auxin. Additionally, the *cis*-elements of phytohormones and stress responses have been identified in many MIR gene promoters [123]. Interestingly, miRNA acts in a non-cell-autonomous manner, and functions as a signal molecule between adjacent tissues even plants [124,125]. Due to their transcriptional regulation by different hormones and their *trans*-regulation capabilities, miRNAs are excellent candidate mediators for the communication between auxin and other hormones.

Interaction between auxin signaling and cytokinin (CK) in plant development has been investigated in recent years. For example, AtARF3 directly activates *ATP/ADP ISOPENTENYLTRANSFERASE 5* (*AtIPT5*) to promote shoot regeneration [126], and AtARF5-mediated *Arabidopsis Response Regulator 5/15* (*AtARR5/15*) repression is important for maintenance of stem cell niches in shoot apical meristems [127]. Moreover, the interdependence among auxin, miRNA and CK has been identified in miR160-AtARF10-AtARR5 module, in which miR160-directed AtARF10 binds to the AtARR15 promoter and inhibits its transcription, thereby promoting CK response during callus initiation [128] (Figure 3). In addition, overexpression of miR160 in legume increases the sensitivity of roots to CK and attenuates expression of CK-regulated transcription factors associated with nodulation, and then inhibits symbiotic nodule development [45]. As protein families which perceive and transduce auxin and CK signaling respectively contain multiple members, it is possible that some members are involved in both auxin- and CK-dependent processes. Hence, a challenge in future would be revealing the complex interaction among auxin, miRNA and CK during plant growth and development.

In addition to CK, the miR160-miR167-ARF module also regulates the expression of several *AtGH3* genes, which, in turn, control auxin and jasmonic acid (JA) homeostasis, and modulate adventitious root initiation [26,28]. The miR167 also controls lateral root development via cleavage of *AtIAR3*, which can hydrolyze IAA conjugates into active IAA and is induced by exogenous JA (Figure 3). These results suggest that miR160 and miR167 function in the crosstalk between auxin and JA during root development through regulating the active IAA and JA contents. Additionally, during flower bud development, the fine-tuned regulation of AtARF6/8, downstream genes of miR167, is essential for the maturation of flowers and JA production [37]; however, the auxin content doesn’t change in developing flower buds, indicating that miR167 might function as a modulator to coordinate auxin signaling and JA biosynthesis [37,55]. Moreover, the miR160-ARF10/16-ABI3 regulatory loop is also essential for seed dormancy, seed germination, and somatic embryogenesis [27,31,32,33]. AtARF10/16 can directly bind to the *AtABI3* promoter and enhance ABA-mediated seed dormancy, while the AtABI3 inhibits the transcription of *AtMIR160b* (Figure 3). Overall, the miR160-miR167-ARF module functions as a potent regulatory node to integrate auxin, CK, ABA and JA signaling pathways. Such a regulation appears in a wide range of developmental processes in plants.

In addition to miR160, miR393 is also a mediator of ABA-to-auxin unidirectional signaling. Under drought stress, the increased ABA promotes the transcription of *MIR393* and represses auxin signaling via the downregulation of *AtTIR1/AFB2* (Figure 3) [65]. In addition, miR393 mediates downregulation of *AtTIR1/AFB2/3* in response to bacterial flagellin (Figure 2C) [63]. Considering the critical role of ABA in biotic stress responses, the ABA-miR393-auxin signaling pathway may be triggered by a bacterial infection via increasing the ABA level [55]. The ABA-miR393-auxin signaling pathway might constitute an important response pathway under biotic and abiotic stresses.

Besides ABA and JA, ethylene (ET) is involved in regulating auxin/miRNAs signaling via inhibiting the transcription of miR156, miR164 and miR390. The miR156-SPL module functions as a hinge to integrate multiple phytohormones, including auxin, gibberellin (GA), and ET, in floral transition and lateral organ development [55]. In rice root, miR156 promotes IAA accumulation via SPL7-mediated transcriptional inhibition of *OsGH3.8* [122]. Interestingly, auxin treatment decreases the accumulation of miR156 in Arabidopsis roots [51], indicating that the auxin-miR156-SPL module might affect root development via precise control of free IAA content. Additionally, miR156 is also involved in GA-dependent floral transition in Arabidopsis [129]. During floral transition, the DELLA protein, a transcriptional repressor of GA signaling, can interact with and repress SPL protein activity [129]. The increased GA induces the degradation of DELLA and activates the SPL protein, subsequently regulating miR156 (Figure 3) [129,130]. Meanwhile, ET regulates GA-dependent DELLA degradation in Arabidopsis to inhibit miR156 transcription in tomato fruit [131,132]. Together, these results suggest that the miR156-SPL module might coordinate GA, ET and auxin for the modulation of the vegetative-phase transition and lateral organ development in response to different environmental conditions.

The expression of miR164 is gradually repressed by AtEIN2, which subsequently de-represses *AtNAC2* during leaf aging [133]. Moreover, the decreased JA level could restore the ET sensitivity of *ein2* and *coi1* (a JA receptor mutant) mutants, suggesting the indirect regulation of miR164 by JA [133,134]. Unlike the miRNA mentioned above, miR390 is only slightly downregulated by ET in *Medicago* root, suggesting that miR390 might participate in the crosstalk of different hormones via the downregulation of its target genes, ARF2/3/4 [55].

With the growing number of studies on the interaction of miRNA and hormone signaling, the number of known target genes of miRNAs involved in different hormone signaling continues to increase. In addition to the hormones described above, new discoveries of miRNA-mediated interactions between auxin and other hormones, including brassinosteroid (BR), CK, salicylic acid (SA), and strigolactones, will deepen our understanding of the crosstalk between auxin and other hormones and make possible the application of miRNAs for crop improvement.

## 4. Concluding Remarks

In plants, auxin plays a central role in growth and development as well as responses to abiotic and biotic stresses [2,3,135,136,137]. Precise regulation of auxin homeostasis and its signaling is extremely critical. In the last two decades, different miRNAs have emerged as master modulators of the auxin response pathway by affecting its biosynthesis, metabolism, distribution, and perception [138].

The control of auxin responses by miRNAs occurs mostly downstream of the TIR1/AFBs by regulating the expression of ARFs and AUX/IAAs, therefore affecting only part of the auxin-regulated processes. Interestingly, auxin and miRNAs independently regulate the target ARF levels when controlling the development of aerial organs and LRs. In addition, some miRNA families (e.g., miR167 and miR393) have been found to act upstream by regulating genes involved in auxin biosynthesis, metabolism and perception. Therefore such miRNAs play a more crucial role in controlling the auxin response. Hence, miRNA-auxin signaling regulatory modules provide additional layers of modulation of auxin signaling. Moreover, miR390-derived tasiARFs move intercellularly to create a small RNA gradient, which can pattern the abaxial determinant *AtARF3* during leaf development [139]. Therefore, deeper investigation of the function of miRNAs will enhance our understanding of the precise regulation of auxin signaling during plant growth and development.

Although there have been great advances in the study of miRNA-mediated auxin signaling, some questions still need to be further investigated:(i)How do plants integrate all these developmental signals or stress responses to control the levels of the individual miRNAs-auxin signaling module?

As mentioned above, we have known that plants employ different miRNAs and then their targets in response to different developmental timing and various stress signals; however, it is still unknown how plants precisely control this process. A deeper understanding of the regulatory mechanism(s) will allow us to choose appropriate targets for modulating agriculturally important traits.

(ii)Which lineage-, species- and tissue-specific miRNA molecules are also involved in regulating auxin responses?

As different environments shape different plants, the plants will evolve the specific miRNAs in response; however, previous studies have mainly focused on the functions of conserved miRNAs. Therefore, it would be valuable to evaluate the role of lineage-, species- and tissue-specific miRNAs to complement our view and to determine how broadly miRNA-dependent genetic networks could influence the auxin signaling in different species.

(iii)How can we use miRNA-auxin modules to create tolerance to stresses and/or high-yield crops?

Because of the extensive regulation of auxin signaling by miRNAs during plant growth and in response to stresses, miRNAs have been considered as good targets for the genetic improvement of crops [140]. Generally, a given miRNA has different roles due to its multiple targets. Hence, it is necessary to make sure that only the target trait (but not other important traits) could be modified before it can be used for plant improvement. Some strategies have been applied to transgenic experiments, including artificial miRNAs, artificial tasiRNAs, artificial target mimics, and overexpression of miRNA-resistant targets. Additionally, CRISPR-Cas9 technology has been used for improvement of crop agronomic traits [141,142,143,144,145,146]. Undoubtedly, with the development of transgenic technologies, miRNA-based breeding strategies will offer a wide range of application scenarios.

## Figures and Tables

**Figure 1 ijms-23-00510-f001:**
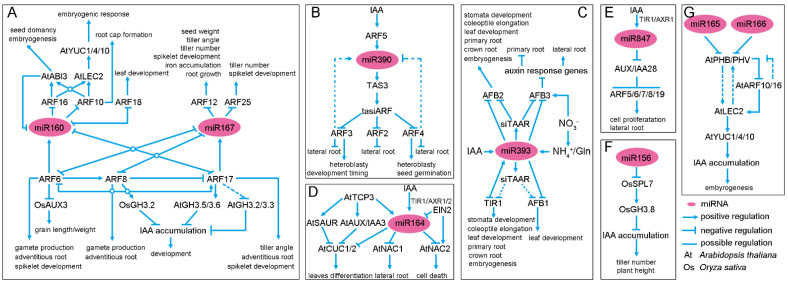
Overview of miRNA-mediated regulatory modules of plant growth and development in Arabidopsis and rice. (**A**–**G**) summary of the molecular connections between specific miRNAs and components of the auxin signaling pathway: (**A**) miR160 and miR167; (**B**) miR390; (**C**) miR393; (**D**) miR164; (**E**) miR847; (**F**) miR156, and (**G**) miR165 and miR166. Abbreviation: TIR1, Transport Inhibitor Response 1; AFB, Auxin F-box Protein; ARF, Auxin Response Factor; LEC2, Leafy Cotyledon 2; AUX/IAA, Auxin/Indole Acetic Acid; ABI3, ABA Insensitive 3; AUX3, Auxin Resistant 3; GH3, Gretchen Hagen 3; TAS3, Trans-Acting SiRNA 3; tasiARF, trans-acting small interfering ARF; TCP3, Teosinte branched/Cycloidea/PCF Transcription Factor 3; SAUR, Small Auxin-Up RNA; CUC, Cup-Shaped Cotyledon; NAC, NAC Domain-Containing protein; EIN2, Ethylene Insensitive 2; AXR1/2, Auxin Resistant 1/2; NH_4_^+^, ammonium; NO_3_^−^, nitrite; Gln, glutamine; siTAAR, small interfering RNA of AFB2/3; SPL7, Squamosal Promoter binding protein-Like 7; PHB/PHV, PHABULOSA/PHAVOLUTA; YUC, Flavin-binding monooxygenase family protein.

**Figure 2 ijms-23-00510-f002:**
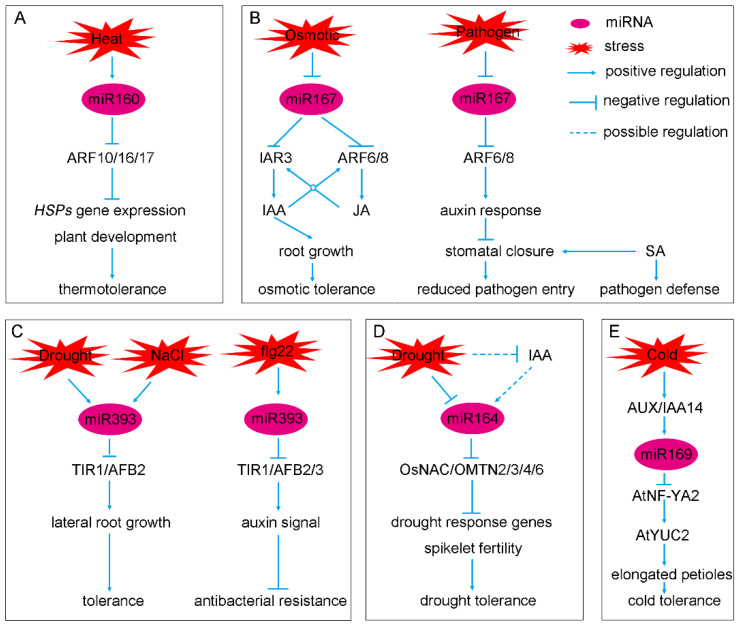
miRNA-auxin signal modules are involved in response to abiotic and biotic stresses in Arabidopsis and rice. (**A**–**E**) Summary of specific miRNAs in response to various environmental stresses: (**A**) miR160; (**B**) miR167; (**C**) miR393; (**D**) miR164, and (**E**) miR169. Abbreviation: HSPs, heat shock proteins; ARF, auxin response factor; IAR3, Indole acetic acid Alanine-Resistant 3; NAC/OMTN, NAC Domain Containing protein in rice; AUX/IAA, Auxin/Indole Acetic Acid; NF-YA2, Nuclear Factor Y subunit A; YUC, Flavin-binding monooxygenase family protein; TIR1/AFB, Transport Inhibitor Response 1/Auxin Signaling F-box Protein; JA, Jasmonic Acid; SA, Salicylic Acid; flg22, 22-amino acid peptide from the N-terminus of eubacterial flagellin.

**Figure 3 ijms-23-00510-f003:**
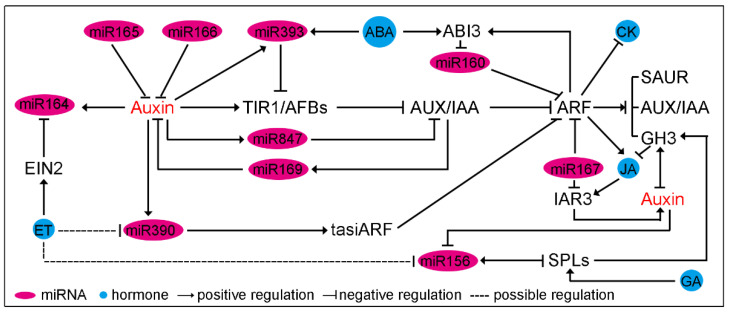
The central role of miRNA-mediated regulation of the crosstalk between auxin and other hormones. Abbreviation: ABA, abscisic acid; CK, cytokinin; ET, ethylene; GA, gibberellin; JA, jasmonic acid.

**Table 1 ijms-23-00510-t001:** Primary/secondary miRNA targets involved in the regulation of auxin signaling in other species.

miRNA	Species	Direct Targets	Secondary Targets	Target Functions	References
miR160	*Solanum lycopersicum*	*SlARF10/17*		Ovary patterning; floral organ abscission; lamina outgrowth; leaf water loss; pathogen defense	[74,75,76,77,78,79,80]
	*Glycine max* L.	*GmARF*		Nodule development	[45]
	*Hordeum vulgare*	*HvARF13/17*		Heat stress	[81]
	*Triticum aestivum* L.	*TaARF*		Abiotic stress	[82]
	*Zea mays* L.	*ZmARF*		Drought stress	[83]
	*Solanum tuberosum*	*StARF10*	*StGH3.6*	Pathogen defense; root architecture	[84,85]
	*Gossypium hirsutum*	*GhARF10/17*		Heat stress; salt stress	[86,87]
	*Dimocarpus longan*	*DlARF10/16/17*		Somatic embryogenesis	[88]
	*Cucumis melo*	*CmARF10/16/17*		Aphid resistance	[89]
	*Medicago truncatula*	*MeARF10/16/17*		Nodule number and development	[90]
	*Manihot esculenta Crantz*	*MaARF10*		Pathogen defense	[91]
	*Prunus persica* L.	*PpARF17*		Fruit enlargement	[92]
	*Dendrobium huoshanense*	*Dhu-20362/-11346*		Drought stress	[93]
miR164	*Solanum lycopersicum*	*SlNAC1*	NA	Pathogen defense	[79,80,94]
miR166	*Larix leptolepis*	*LaHDZIPIII*	*LaNIT*	Somatic embryogenesis	[95]
miR167	*Solanum lycopersicum*	*SlARF6/8*		Pathogen defence	[75,79,96,97]
	*Glycine max*	*GmARF8a/8b*		Nodulation; lateral root development	[46]
	*Cucumis melo*	*CmARF6/8*		Aphid resistance	[89]
	*Hordeum vulgare*	*HvARF8*		Heat stress	[81]
	*Populus* spp.	*PeARF8*		Adventitious rooting	[98]
	*Nicotiana tabacum* L.	*NtARF6/8*		Phosphorus starvation	[99]
miR390	*Zea mays*	NA	NA	Drought stress	[83]
	*Dimocarpus longan*	*DlTAS3*	*tasiDlARF3/ARF4*	Somatic embryogenesis	[88]
	*Cucumis melo*	NA	*tasiCmARF*	Aphids resistance	[89]
	*Medicago truncatula*	*MeTAS3*	*tasiMeARF2/3/4*	Nodulation; lateral root growth	[57]
	*Mimulus lewisii*	*MlTAS3*	*tasiMlARF3/4*	Corolla tube formation	[100]
	*Physcomitrella patens*	*PpTAS3*	*tasiPpARF*	Developmental timing	[101]
	*Helianthus tuberosus* L.	*HtTAS3a/b/c*	*tasiHtARF1/2/3*	Salt stress	[58]
	*B. oleracea* L. *var. italica*	*BoTAS3*	*tasiBoARF2/3/4*	Lateral root	[102]
	*Populus* spp.	NA	*tasiPsARF3.1/3.2/4*	Salt stress	[103]
	*Nicotiana tabacum* L.	NA	*tasiNtARF*	*Manduca sexta* resistance	[104]
miR393	*Glycine max*	*GmTIR1/AFB3*		Nodulation	[105]
	*Hordeum vulgare*	*HvTIR1/AFB*		Aluminum stress	[106]
	*Zea mays*	*ZmTIR1/AFB*		Drought stress	[83]
	*Cucumis sativus* L.	*CsTIR1/AFB2*		Fruit/seed set development and leaf morphogenesis	[107]
	*Cucumis melo*	*TIR1/AFB2*		Aphids resistance	[89]
	*Manihot esculenta* Crantz	*MaTIR1*		Pathogen defence	[91]
	*Prunus persica* L.	*PpAFB2*		Fruit enlargement	[92]
	*Malus* × *domestica* Borkh.	*MdTIR1A*		Adventitious root formation	[108]
	*Panicum virgatum* L.	*PvTIR1/AFB1/2/3*		Cold stress; tillering	[109]
miR1153-y	*Triticum aestivum.*	*TaGH3.*7		Post-mowing regeneration	[110]
miRN43	*Litchi chinensis* Sonn.	*LcARF9*		Fruit abscission	[111]

NA: information not available.

## Data Availability

Not applicable.

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
