# Peer review of "MicroRNAs Are Involved in Regulating Plant Development and Stress Response through Fine-Tuning of TIR1/AFB-Dependent Auxin Signaling"

_ijms, 2022, doi:10.3390/ijms23010510_

Round 1
Reviewer 1 Report
Micro RNA mediated auxin regulation is one of the most well studied mechanisms in plants, particularly in the reference genome Arabidopsis and rice. Although, this article provided information on non-model plants, it failed to underscore the commonalities and divergences in terms of their functionality and target module. Secondly, the article has been written in a very descriptive way without deducing major conclusions, interpretations and future directions, which are expected from this kind of a review article. In the title the authors promised to provide information on the role of micro RNAs in plant development and stress responses. However, the main manuscript is quite intermingled and it is hard to get information about micro RNAs involved in different abiotic stress (e.g., salt, drought, cold, high temperature), biotic stress and plant development (e.g., leaf, root, flower development). There should also be section that discusses cross talk between stress response and plant development. It is also important to discuss current challenges and future prospects to develop stress resilient crops using these knowledge. The provided Figures are not innovative one and are repetitive of the information provided in the text.
Author Response
Reviewer 1#
Micro RNA mediated auxin regulation is one of the most well studied mechanisms in plants, particularly in the reference genome Arabidopsis and rice. Although, this article provided information on non-model plants, it failed to underscore the commonalities and divergences in terms of their functionality and target module.
We have carefully considered your comments. To further underscore the commonalities and divergences of a special miRNA in Arabidopsis, rice and non-model plants, we have performed the following revisions:
(1) We have presented the subtitle 2 and subtitle 3 together, the new subtitle 2 renamed as “The crosstalk of auxin, miRNA and other hormones during plant development and stress responses”, and reorganized the role of a special miRNA across different species in the new manuscript. In addition, we also invited a native English speaker to edit our manuscript, which could enhance the readability of the new manuscript.
2) We also reorganized the Table 1, in which grouped the species that present a common miRNA, making the correspondence to the targets and function.
Secondly, the article has been written in a very descriptive way without deducing major conclusions, interpretations and future directions, which are expected from this kind of a review article. In the title the authors promised to provide information on the role of micro RNAs in plant development and stress responses. However, the main manuscript is quite intermingled and it is hard to get information about micro RNAs involved in different abiotic stress (e.g., salt, drought, cold, high temperature), biotic stress and plant development (e.g., leaf, root, flower development). There should also be section that discusses cross talk between stress response and plant development.
We thank the reviewer for the suggestion, and we have made the following revisions in response to your comments:
(1) We added more conclusions, interpretations and future directions in the new manuscript, which described as:
“However, the environmental or developmental signals and the mechanisms underlying the transition of seed dormancy/germination still need to be further investigated.”
“It would be interesting to investigate which ARFs are employed to control the expression of miR390 during different developmental contexts.”
“Additionally, the roles of miR393 in regulating both normal plant developmental processes, such as nodulation, adventitious root formation, leaf morphogenesis and fruit development, and the response to different environmental stresses, including aluminum stress, drought stress, cold stress and aphid infection, have been widely studied in different species (Table 1). However, due to lack of knockout mutant of MIR393, the exact roles miR393 in plant development and stress responses still needs more investigation.”
……
(2) We provided more description about miRNA involved in different stresses and plant development in different species in the last paragraph of every subtitle.
(3) Furthermore, we also added a new part which discussed the role of miRNA in regulating stress responses and plant development via integrating the different hormones. See the Subtitle 3: The crosstalk of auxin, miRNA and other hormones during plant development and stress responses.
It is also important to discuss current challenges and future prospects to develop stress resilient crops using these knowledge. The provided Figures are not innovative one and are repetitive of the information provided in the text.
We thank the reviewer for the comments. The advice is correct and appreciated. We have discussed more in the manuscript, now described as:
“iii) How can we use miRNA-auxin modules to create tolerance to stresses and/or high-yield crops?
Because of the extensive regulation of miRNAs during plant growth and in response to stresses, miRNAs have been considered as good targets for the genetic improvement of crops [140]. Generally, a given miRNA has different roles due to its multiple targets. Hence, it is necessary to make sure that only the target trait (but not other important traits) could be modified before it can be used for plant improvement. Some strategies have been applied to transgenic experiments, including artificial miRNAs, artificial tasiRNAs, artificial target mimics, and overexpression of miRNA-resistant targets. Additionally, CRISPR-Cas9 technology has been used for improvement of crop agronomic traits [141-146]. Undoubtedly, with the developmental of transgenic technologies, miRNA-based breeding strategies will offer a wide range of application scenarios.”
In addition, we also added a new Figure (Figure 3) in the manuscript to provide more information about the role of miRNA during the crosstalk between auxin and other hormones.
Reviewer 2 Report
ijms-529857-peer-review-v1
Full Title: MicroRNAs are Involved in Regulating Plant Development and Stress Response through Fine-Tunning of TIR1/AFBs-dependent Auxin Signaling
General comments to the manuscript:
The present manuscript aims to integrate the main research focused on miRNAs as a regulatory mechanism dependent on auxin signaling, with a key role in plant development and stress response. The topic of the manuscript is of high interest for the readers of the International Journal of Molecular Sciences journal; however, important flags were identified which make it not possible to accept it for publication in its present form. Overall, the language used is not of enough quality, there are several grammar mistakes, and confusing sentences (e.g. “However, the lack of knockout mutant of OsMIR393a and OsMIR393b results in the exact role of miR393 in rice development and stress response is still unclear”; “While, the induction of AtMIR164 by exogenous auxin is also disappeared in the mutants auxin resistant 1/2 (axr1/2) and Attir1 (73)”), in some cases associated to wrong ideas (please see the example in page 4 “to defend pathogen” must be changed by “to defend the plant against the pathogen”).
Besides the language, the manuscript must be carefully revised since there are some incomplete sentences. There are examples “that contribute to the diversity of auxins (8)” being not clear if the diversity is in terms of function. Another example “And, the transcription is regulated …” but it is not clear if the authors are referring to TIR/AFBs or Aux/IAA genes. “The recent study” but is not clear to each study are the authors referring to. Many examples turn not possible to fully understand the idea that the authors want to transmit.
In the introduction I suggest authors better establish the link between miRNAs and auxin signaling. It seems that the authors lost the focus after the general description of miRNAs. The interest of SEAP1 in the context of the manuscript is not clearly explained.
I suggest presenting the point 3 and point 2 together, removing from the title of point 2 “in Arabidopsis and rice”. In my opinion, makes more sense to present the role of a specific miRNA across different species.
References to Figures must be included in the text.
Regarding Table 1, to avoid repetitions and to make easier the comparison across the different species, I suggest grouping the species that present a common miRNA, making the correspondence to the targets and function.
Author Response
Reviewer 2#
The present manuscript aims to integrate the main research focused on miRNAs as a regulatory mechanism dependent on auxin signaling, with a key role in plant development and stress response. The topic of the manuscript is of high interest for the readers of the International Journal of Molecular Sciences journal; however, important flags were identified which make it not possible to accept it for publication in its present form. Overall, the language used is not of enough quality, there are several grammar mistakes, and confusing sentences (e.g. “However, the lack of knockout mutant of OsMIR393a and OsMIR393b results in the exact role of miR393 in rice development and stress response is still unclear”; “While, the induction of AtMIR164 by exogenous auxin is also disappeared in the mutants auxin resistant 1/2 (axr1/2) and Attir1 (73)”), in some cases associated to wrong ideas (please see the example in page 4 “to defend pathogen” must be changed by “to defend the plant against the pathogen”).
We have carefully considered your comments, and have invited a native English speaker to edit our manuscript.
Besides the language, the manuscript must be carefully revised since there are some incomplete sentences. There are examples “that contribute to the diversity of auxins (8)” being not clear if the diversity is in terms of function. Another example “And, the transcription is regulated …” but it is not clear if the authors are referring to TIR/AFBs or Aux/IAA genes. “The recent study” but is not clear to each study are the authors referring to. Many examples turn not possible to fully understand the idea that the authors want to transmit.
We are so sorry about our poor English writing, which causes a variety of errors and creates confusion and miscommunication. We have carefully considered your comments and have revised all the major and minor errors that you pointed out. In addition, we also have considerable revised the manuscript in the help of a professional service.
In the introduction I suggest authors better establish the link between miRNAs and auxin signaling. It seems that the authors lost the focus after the general description of miRNAs. The interest of SEAP1 in the context of the manuscript is not clearly explained.
Thanks for your suggestion, we carefully reorganized this paragraph, it is described like this:
“In addition to the precise regulation of auxin biosynthesis and distribution, the auxin signaling pathway is subject to locus-specific control by microRNAs (miRNA) [16, 17]. miRNAs are short, single-stranded nucleic acid molecules (~21-24 nucleotides) that repress target gene expression as an additional layer of regulation of plant growth and stress responses in both plants and animals [18]. In plants, primary miRNAs (pri-miRNAs) are endogenously expressed from miRNA-encoding genes (MIRs) by the DNA-dependent RNA polymerase II (Pol II) [19]. The pri-miRNA is cleaved by RNase â…¢ enzymes Dicer-Like 1(DCL1)/DCL4 and then processed into a mature miRNA by a protein complex composed of SERRATE (SE), HYPONASTIC LEAVES 1 (HY1), TOUCH and DCL1 [20]. miRNA biogenesis is also regulated by various protein factors at the transcriptional and posttranscriptional levels [21]. A recent study reports a new protein factor, SE-Associated Protein 1 (SEAP1), that associates with the DCL1 complex and promotes the interaction of the DCL1 complexes with pri-miRNAs, facilitating pri-miRNA splicing, processing, and/or stability [21]. miRNAs regulate gene expression by directly cleaving target mRNA or inhibiting translation in the cytoplasm through conserved ARGONAUTE (AGO) proteins [20]. In addition, miRNAs can bind to target promoters in the nucleus, thereby activating expression [22], suggesting that miRNAs can function as a gene activator or repressor. In plants, miRNAs exhibit high evolutionarily conserved complementarity to mRNAs among dicot and monocot species [23]. Conserved miRNAs may play important roles in basic development and signal transduction, while non-conserved miRNAs may play particular roles in different tissues or plant species under special environment conditions [24-26].”
I suggest presenting the point 3 and point 2 together, removing from the title of point 2 “in Arabidopsis and rice”. In my opinion, makes more sense to present the role of a specific miRNA across different species.
Thanks for your insightful and constructive suggestions. We have presented the point 2 and point 3 together, the new point 2 renamed as “miRNA-mediated regulation of auxin homeostasis and signaling”. In addition, we have reorganized the role of a special miRNA across different species in the new manuscript.
References to Figures must be included in the text.
Done as suggested.
Regarding Table 1, to avoid repetitions and to make easier the comparison across the different species, I suggest grouping the species that present a common miRNA, making the correspondence to the targets and function.
Thanks for your suggestion, we carefully reorganized the Table 1 as you suggested. The new Table 1 would be more accurate.
Reviewer 3 Report
The manuscript is a summary of latest studies dedicated to miRNA- mediated regulation of key components of auxin signaling, mainly in rice and Arabidopsis.
Whereas the topic microRNA/hormones is very complex as it involves many aspects of the plant physiology, the risk is to make only a nice summary without making further real critical perspectives.
The quality of the manuscript is poor, the figures are unclear, not at all described and do not add anything to the manuscript.
In the text when referring to a figure it is never indicated which part of the figure is being referred to (A,B,C,D,E,F,G?)
Table 1 is inserted in the text but never commented on. It is not clear what its meaning is
In the abstract the relationship with other hormones is mentioned but little or nothing discussed in the text
Author Response
Reviewer 3#
The manuscript is a summary of latest studies dedicated to miRNA- mediated regulation of key components of auxin signaling, mainly in rice and Arabidopsis.
Whereas the topic microRNA/hormones is very complex as it involves many aspects of the plant physiology, the risk is to make only a nice summary without making further real critical perspectives.
We thank the reviewer for the comments. The advice is correct and appreciated. We did our best to add more discussions and perspectives in the new manuscript, which described as:
“However, the environmental or developmental signals and the mechanisms underlying the transition of seed dormancy/germination still need to be further investigated.”
“It would be interesting to investigate which ARFs are employed to control the expression of miR390 during different developmental contexts.”
“Additionally, the roles of miR393 in regulating both normal plant developmental processes, such as nodulation, adventitious root formation, leaf morphogenesis and fruit development, and the response to different environmental stresses, including aluminum stress, drought stress, cold stress and aphid infection, have been widely studied in different species (Table 1). However, due to lack of knockout mutant of MIR393, the exact roles miR393 in plant development and stress responses still needs more investigation.”
……
The quality of the manuscript is poor, the figures are unclear, not at all described and do not add anything to the manuscript.
In the text when referring to a figure it is never indicated which part of the figure is being referred to (A,B,C,D,E,F,G?)
We are so sorry about our poor English writing, and have invited a native English speaker to edit our manuscript. In addition, we have added more description of figures in the Figure legends, and inserted the Figure (A, B, C, D, E, G) to the desired position in the manuscript.
Table 1 is inserted in the text but never commented on. It is not clear what its meaning is
Yes, this is a label-missing, we apologize for the error. We have corrected this error in the new manuscript.
In the abstract the relationship with other hormones is mentioned but little or nothing discussed in the text
We thank the reviewer’s comments. We have added a new part to discuss these relationships in the manuscript, now described as:
“3. The crosstalk of auxin, miRNA and other hormones during plant development and stress responses
Auxin controls most developmental stages and stress responses in plants, and the communication between auxin and the other phytohormones (ethylene, abscisic acid, jasmonic acid, etc.) is critical to these processes. Although it is clear that there is phytohormone crosstalk, only a few mediators have been investigated thus far, including miRNAs [59].
…….
With the growing number of studies on the interaction of miRNA and hormone signaling, the number of known target genes of miRNAs involved in different hormone signaling continue to increase. In addition to the hormones described above, new discoveries of miRNA-mediated interactions between auxin and other hormones, including brassinosteroids (BRs), cytokinins (CKs), salicylic acid (SA), and strigolactones, will deepen our understanding of the crosstalk between auxin and other hormones and make possible the application of miRNAs for crop improvement.”
In addition, we also added a new figure (Figure 3) to summarize the relationship of auxin-miRNA-other hormones.
Round 2
Reviewer 2 Report
I would like to thank the authors for the effort made to improve the manuscript.
Author Response
We gratefully appreciate for your recognition of our revised manuscript.
Reviewer 3 Report
The authors have sufficiently replied to the comments
Author Response
Thank you very much for your recognition of our revised manuscript.